# A 2D material–based transparent hydrogel with engineerable interference colours

Baofu Ding[1,4], Pengyuan Zeng[1,4], Ziyang Huang[1], Lixin Dai[1], Tianshu Lan[1], Hao Xu[1], Yikun Pan[1], Yuting Luo[1], Qiangmin Yu[1], Hui-Ming Cheng [1,2,3] & Bilu Liu [1✉]

Transparent hydrogels are key materials for many applications, such as contact lens, imperceptible soft robotics and invisible wearable devices. Introducing large and engineerable optical anisotropy offers great prospect for endowing them with extra birefringence-based functions and exploiting their applications in see-through flexible polarization optics. However, existing transparent hydrogels suffer from limitation of low and/or non-fine engineerable birefringence. Here, we invent a transparent magneto-birefringence hydrogel with large and finely engineerable optical anisotropy. The large optical anisotropy factor of the embedded magnetic two-dimensional material gives rise to the large magneto-birefringence of the hydrogel in the transparent condition of ultra-low concentration, which is several orders of magnitude larger than usual transparent magnetic hydrogels. High transparency, large and tunable optical anisotropy cooperatively permit the magnetic patterning of interference colours in the hydrogel. The hydrogel also shows mechanochromic and thermochromic property. Our finding provides an entry point for applying hydrogel in optical anisotropy and colour centred fields, with several proof-of-concept applications been demonstrated.

[1] Shenzhen Geim Graphene Center, Tsinghua-Berkeley Shenzhen Institute & Institute of Materials Research, Shenzhen International Graduate School, Tsinghua University, Shenzhen 518055, China. [2] Shenyang National Laboratory for Materials Science, Institute of Metal Research, Chinese Academy of Sciences, Shenyang 110016, China. [3] Advanced Technology Institute, University of Surrey, Guildford, GU 27XH, UK. [4]These authors contributed equally: Baofu Ding and Pengyuan Zeng. ✉email: bilu.liu@sz.tsinghua.edu.cn

A hydrogel is a water-based three-dimensional network of crosslinked polymer chains. Due to the similarity with biological tissues, the hydrogel possesses unparalleled elasticity, irritability, wettability, and biocompatibility, allowing its applications in diverse fields like biomedicine, environmental protection, clean energy, intelligent driving, and smart sensing[1–4]. Recently, great attentions have been paid to the synthesis of transparent hydrogels to extend their applications in see-through electronics[5,6], imperceptible soft robots[7,8], and other optic fields[9–15], such as contact and liquid lens, in vivo optical sensors and fibres for body-contact optics, as well as thermo-, electro-, and humidity-controlled smart windows. Introducing optical anisotropy into widely-used transparent hydrogels will potentially endow them with extra birefringence-based functions[16,17], including digital coding, chemical sensing, polarization navigation, disease diagnosis, etc. Remarkably, the large birefringence can make a transparent hydrogel appear transmitted interference colours. Contrary to the colours from dyes or pigments, interference colours show unique vivid, metallic, wide colour gamut and non-photobleaching advantages[16–20]. Therefore, it is highly desired to introduce interference colour functions into hydrogels. Unfortunately, current reported transparent hydrogels can only display the black-to-white switch and is not interference colourful. With this regard, large and tuneable optical anisotropy is a prerequisite to extend the applications of transparent hydrogel to the optical-anisotropy and colour-related fields.

Moreover, applications of hydrogel in anisotropic optics impose harsh requirements on the uniformity and controllability of anisotropic structure. The conventional approaches to fabricating anisotropic hydrogel include preparation of aligned functional units by means of self-assembly, exertion of compressing, stretching, or shearing forces, electric fields, and 3D printing[21–24]. These methods usually suffer shortcomings of uneven distribution of force and electric field which make the uniformity and precise tunability of structural anisotropy difficult, and consequently limit programmable control of spatially resolved optical anisotropy[24,25]. In contrast, the magnetic field can be finely tuned, and appears homogeneous in a sizable area[23,26]. Alongside the non-invasive and contact-free advantages, magnetic field control is envisaged to be a promising tool for achieving engineerable anisotropy in an accurately tuneable and uniform manner[26–28]. In spite of this, the usual magnetically responsive functional units such as iron oxide nanoparticles, possess one or both of the following drawbacks, i.e., strong optical absorption and low optical anisotropy. These two features severely limit the length of optical path $L$ and birefringence $\triangle n$ of the system. As a result, the condition for the realization of transmitted interference colour, namely, $\triangle nL > 400$ nm cannot be satisfied in these systems according to the well-known Michel–Lévy chart. Considering the interplay between the shape anisotropy and optical/magnetic anisotropy[18,29–32], magnetic two-dimensional (2D) materials with a wide optical bandgap offer excellent prospects for realization of a transparent hydrogel with engineerable interference colours, as their shape anisotropy shows two or three orders of magnitude larger than those of other dimensional systems.

Here, we report the invention of a magneto-birefringence-based transparent hydrogel (MB-hydrogel), fabricated by using a 2D paramagnetic material of cobalt-doped titanium oxide (CTO) as a functional unit. The embedded CTO has a large optical anisotropy factor of $2.85 \times 10^{-11}$ $C^2 J^{-1} m^{-1}$, which is at least one order of magnitude larger than the highest value in other nanomaterials. Such hydrogel demonstrates sensitive magnetic response, large and uniform optical anisotropy as well as the resultant multiple transmitted interference colours. Peak wavelength for each colour is tuned via magnetically controlled alignment of 2D CTO materials during hydrogelation. Based on the obtained optical properties, the room-temperature solution-processable MB-hydrogel shows great potential for personalized optical applications, such as the optical phase retarder, gradient optical attenuator, magnetic see-through colour imager, and mechano-chromic indicator.

## Results

### Synthesis and characterization of the MB-hydrogel with aligned 2D CTO.
Synthesis of MB-hydrogel is schemed in Fig. 1a. Briefly, the processes include the preparation of 2D CTO aqueous suspension, where 2D CTO materials were exfoliated by using a four-stage method from a layered lepidocrocite-type structure bulk[18] (Methods, Supplementary Fig. 1). Shape anisotropy of CTO is described by its aspect ratio of $l/t$, where $l$ and $t$ are average lateral size and thickness of the flake (Fig. 1a), having average values of 1.8 μm and ~1.3 nm, respectively (Supplementary Fig. 2). The ratio is one to two orders of magnitude larger than those of other-dimensional materials, such as nanorods[33] and nanoplates[34]. Then, ultraviolet (UV) curable MB-resin was prepared by adding a monomer of poly(ethylene glycol) diacrylate and photo-initiator of 2-Hydroxy-4′-(2-hydroxyethoxy)-2-methylpropiophenone to the CTO suspension. Whereafter, hydrogelation of the MB-resin was performed in the presence of a magnetic field **H** and UV illumination (Methods). Based on XPS measurements (Supplementary Fig. 3), we detected a peak shift of Ti $2p_{3/2}$ from 458. 47 eV (pure 2D CTO flake, which is in good agreement with the standard value of 458.50 eV for Ti $2p_{3/2}$ of $TiO_2$) to lower binding energy of 457.44 eV (polymer-PEGDA modified 2D CTO flakes). Such peak shift indicates interface interaction between CTO and polymers, presumably due to the formation of hydrogen bond between hydrogen in a polymer network and oxygen in CTO, making the $Ti^{+4}$ to $Ti^{+(4-\delta)}$. When we compare Ti-O-H (surface –OH) with Ti-O (surface –O), the O in Ti-O-H can withdraw electrons from both Ti and H due to its large electronegativity, while O in Ti-O can only withdraw electrons from Ti. This leads to a less decrease of $3d/4s$-orbital electron density of Ti in Ti-O-H than in the Ti-O. Such less decrease of Ti $3d/4s$- electron density will result in an outward shift of Ti $2p$-orbital electrons in Ti-O-H than Ti-O, due to the so-called screening effect. As a result, this shift leads to smaller binding energy for Ti $2p$ in Ti-O-H than in Ti-O case. This result is also in accordance with a previous similar analysis of M-O-H bonding, where M is the typical metal[35].

If the CTO-hydrogel interface contains hydrogen bond, it can be envisioned that the introduction of this interface can increase the mechanical strength and durability of the hydrogel. To verify this point, we characterized the mechanic properties of hydrogels without and with CTO flakes. We found that the addition of CTO can remarkably improve the mechanical performance of hydrogel as evidenced by the measured tension/compression-strain curves (Supplementary Fig. 4) as well as the mechanic durability (Supplementary Fig. 5). For instance, for tension-strain correspondence, the stress of hydrogel at 20% increases from 10 kPa to 15 kPa by adding ordered CTO functional units, giving rise to an increase by 50 %. For durability, the retention rate of hydrogel without CTO significantly drops to 60% after 14 times cycling. Meanwhile, the bare hydrogel was fractured into several pieces, indicating poor durability. While after adding CTO, the hydrogel shows much-improved durability, as evidenced by the retention rate of >90%, even after 50-time cycling.

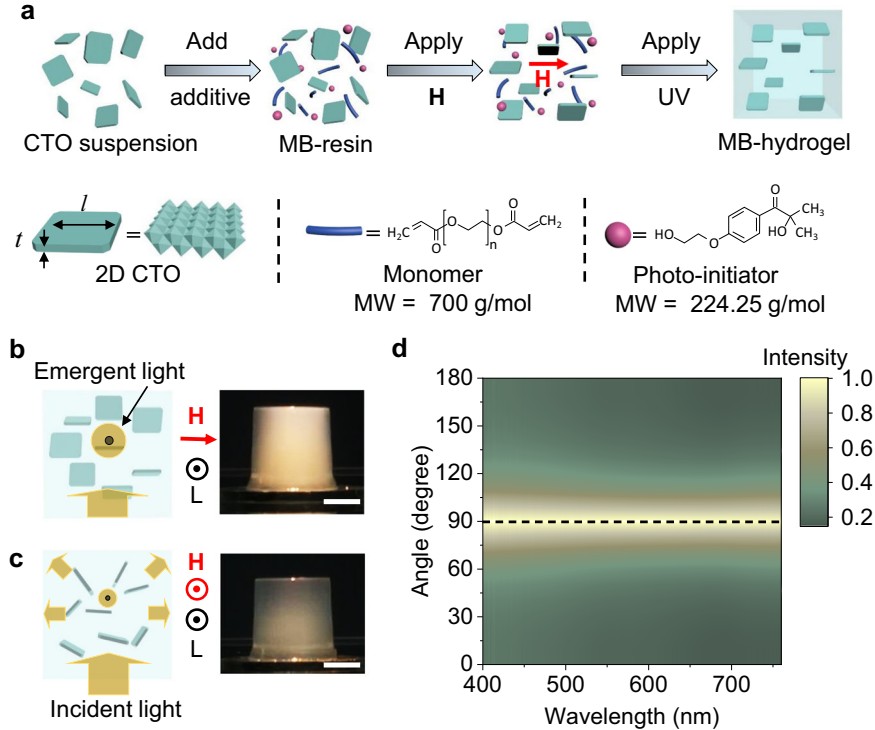

**Fig. 1 Fabrication of magneto-birefringence hydrogel (MB-hydrogel). a** Fabrication process of the MB-hydrogel. The MB-resin is a mixed suspension, comprising 2D Co-doped TiO$_2$ (CTO) materials, water, monomer (PEGDA, molecular weight MW of 700 g/mol), and photo-initiator (Irgacure 2959, MW of 224.25 g/mol). Shape anisotropy of an individual 2D CTO is presented by the thickness $t$ and the lateral size $l$. **b, c** Schemes (left) and images (right) of light scattering from MB-hydrogel-2 for L ⊥ **H**-axis (**b**) and L∥**H**-axis (**c**). L represents emergent light. **d** The normalized intensity of scattered light from the MB-hydrogel at different angles between L and **H**-axis. All scale bars, 5 mm.

The magnetic field direction is defined as **H**-axis for the MB-hydrogel even after the field removal. The as-fabricated MB-hydrogel-1 possesses high transmittance (>90% in the visible spectral region, Supplementary Fig. 6), uniform optical anisotropy (<1% variation rate in a 6 mm × 6 mm region, Supplementary Fig. 7), and high elasticity (Supplementary Movie 1). A series of MB-hydrogels with customized optical properties (Supplementary Table S1) were fabricated by synthetically considering three factors, including the shape and thickness ($L$) of hydrogels, the concentration of 2D CTO materials ($C$), and the strength of magnetic field ($\mu_0 H$). With an applied magnetic field, orientations of suspended 2D CTO materials in the MB-resin change from random (zero fields) to ordered ones (at the certain field) as schemed in Fig. 1a. The alignment of the 2D CTO along with **H**-axis can be characterized by the angle-resolved light scattering. The optical setup is illustrated in the left panels of Fig. 1b, c, where light is incident from the bottom of MB-hydrogel-2 and emergent horizontally. Comparison between two snapshots in Fig. 1b and c indicates the enhanced scattering for the emergent light, being perpendicular to **H**-axis (L ⊥ **H**-axis). Angle-resolved scattering mapping result shows that the intensity of scattered light increases with the angle from 0° to 90° (Fig. 1d, Supplementary Movie 2), which is between **H**-axis and the direction of emergent light. The highest intensity is recorded at 90° due to the largest scattering cross-section provided by 2D CTO. In addition, when increasing the concentration of CTO to the value of 0.2 vol%, the perceptible macroscopic domains form in the MB-resin and orient parallel to the field flux with $\mu_0 H = 0.6$ T (Supplementary Fig. 8). Moreover, as shown in scanning electron microscopy (SEM) images (Supplementary Fig. 9), in a freeze-dried hydrogel without a magnetic field, the orientation of the 2D CTO materials in its cross-section presents an irregular distribution. While for the sample with a magnetic

field of 1 T, most of the 2D CTO materials orient along with the external magnetic field. Together with the optical method in Fig. 1b–d, they jointly confirm the parallel alignment of 2D CTO materials with **H**-axis in the MB-hydrogel.

**Optical anisotropy and transmitted interference colour of the MB-hydrogel.** To characterize the optical anisotropy of the MB-hydrogel, we measured output polarization states of transmitted light as a function of azimuth $\theta$, where $\theta$ is between **H**-axis and the polarization vector of incident laser light. The optical setup includes a polarizer, the MB-hydrogel-3 ($\mu_0 H = 220$ mT), and a crossed analyser (Fig. 2a). When back-lit with a 450 nm laser beam, transmitted intensity oscillates with $\theta$ (Fig. 2b), adjusted by rotating the MB-hydrogel-3 along the light direction. Three transmittance minima and two maxima emerge alternately from 0° to 180°. Such sinusoidal-like behaviour complies with the birefringence-based relation, derived from Malus's Law[33],

$$I = I_0 \sin^2(2\theta)\sin^2(\delta/2) \qquad (1)$$

where $\delta = 2\pi \triangle n L/\lambda$. The transmittance minima and maxima correspond to $\theta = (N-1)\pi/4, (N = 1, 2, 3, 4, 5)$. Note that, removal of the analyzer allows the observation of polarization evolution of output light with $\theta$. Upon increasing $\theta$, ellipticity evolves from initial zero ($\chi \approx 0°$ linear polarization at $\theta = 0°$) to the largest value ($\chi \approx 45°$ right circular polarization at $\theta = 45°$), followed by becoming the negligible one at $\theta = 90°$ (Fig. 2c). The $\theta$-dependent intensity as well as polarization test confirms the large optical anisotropy of the MB-hydrogel, in which the **H**-axis serves as the slow axis.

To check the engineerability of optical anisotropy by **H**, we fabricated 9 different MB-hydrogels (No. 4# to 12#) ($L = 2$ mm; $C = 0.02$ vol%), under different $\mu_0 H$, ranging from 0 to 800 mT at an

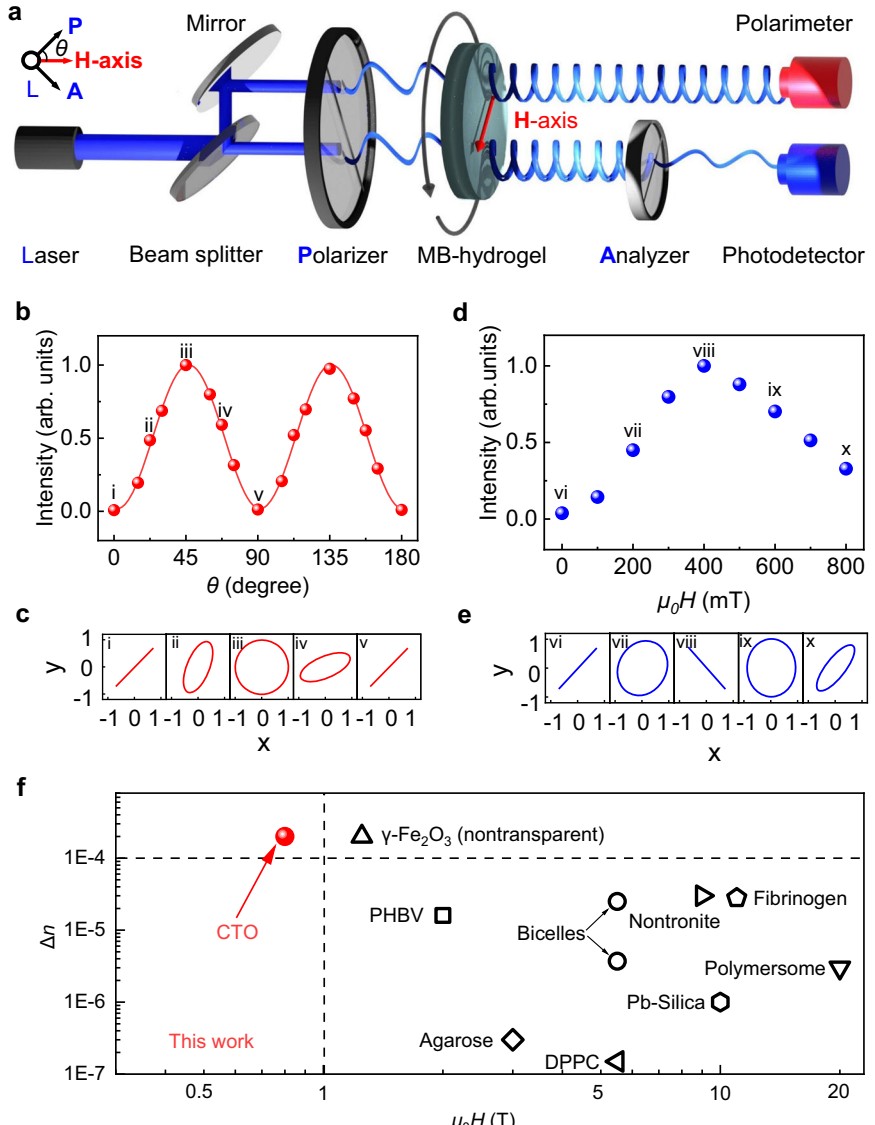

**Fig. 2 Optical anisotropy of the MB-hydrogel. a** Optical setup for measuring polarization and intensity of the transmitted light. **b**, **c** Intensity (**b**) and polarization (**c**) evolution of transmitted light as a function of azimuth $\theta$ relevant to **H**-axis. The field strength and the wavelength are set as 220 mT and 450 nm, respectively. **d**, **e** Intensity (**d**) and polarization (**e**) evolution of transmitted light as a function of **H** strength. The azimuth $\theta$ is set as 45°. **f** Comparison of birefringence $\Delta n$ versus magnetic field for various magnetic hydrogels with the thickness of 10 mm. CTO is 2D cobalt-doped titanium oxide. PHBV, DPPC, and Pb-Silica stand for 1,2-dipalmitoyl-sn-glycero-3-phosphocholine, poly (3-hydroxybutyric acid-co-3-hydroxyvaleric and lead (Pb)-doped silica, respectively. Two dots for Bicelles represent the thulium ion ($Tm^{3+}$) and the dysprosium ion ($Dy^{3+}$)-chelating bicelles. Among these systems, the $\gamma\text{-}Fe_2O_3$ based hydrogel is opaque, while others are transparent for the thickness of 10 mm. Details for data points are shown in Supplementary Table 2.

interval of 100 mT. Maximum intensity was observed for MB-hydrogel-8 ($\mu_0 H = 400$ mT) (Fig. 2d), where the linear polarization state of output light is perpendicular to that of incident light (Fig. 2e). This implies $\delta$ at 400 mT is equal to $\pi$ according to Eq. (1). The identity in $L$ and $\lambda$ for all 9 MB-hydrogels indicates that variation in $\delta$ originates from the magneto-birefringence $\Delta n(H)$, which are calculated to be in the range of 0 to $2.0 \times 10^{-4}$ (Supplementary Fig. 10a). It is worth noting that for MB-hydrogels (No. 10# to 12#), similar to the description of electro-birefringence[36], the magneto-birefringence can be described by $\Delta n = \Delta n_s - A(H - H_0)^{-2}$, where $\Delta n_s$ is the saturate birefringence, $A$ and $H_0$ are two other constant parameters. By using the relation to fit the magneto-birefringence result (Supplementary Fig. 10b), $\Delta n_s$ is yielded to be about $2.4 \times 10^{-4}$. As a result, the intrinsic optical anisotropy factor $\Delta g$ of 2D CTO materials, which is independent of

the concentration, can be determined and estimated to be $2.85 \times 10^{-11}$ $C^2 J^{-1} m^{-1}$, according to $\Delta g = \frac{2n\varepsilon_0}{C_{vol}} \Delta n_s$, where $n$, $\varepsilon_0$, and $C_{vol}$ are average suspension refractive index, vacuum dielectric constant and volume concentration of 2D material, respectively. Such factor is more than one order of magnitude larger than the highest reported value in 2D materials ($1.62 \times 10^{-12}$ $C^2 J^{-1} m^{-1}$)[36,37], and even two-times larger than the tolane-based highest-birefringence liquid crystal, such as bitolane, phenyl tolane[38].

The large factor of 2D CTO material enables the development of a transparent hydrogel with a record-high magneto-birefringence, which is more than one order of magnitude larger than all previously known transparent magnetic hydrogels (Fig. 2f), while the required strength of magnetic field is drastically decreased from several tesla to the value of <1 tesla. For example, these

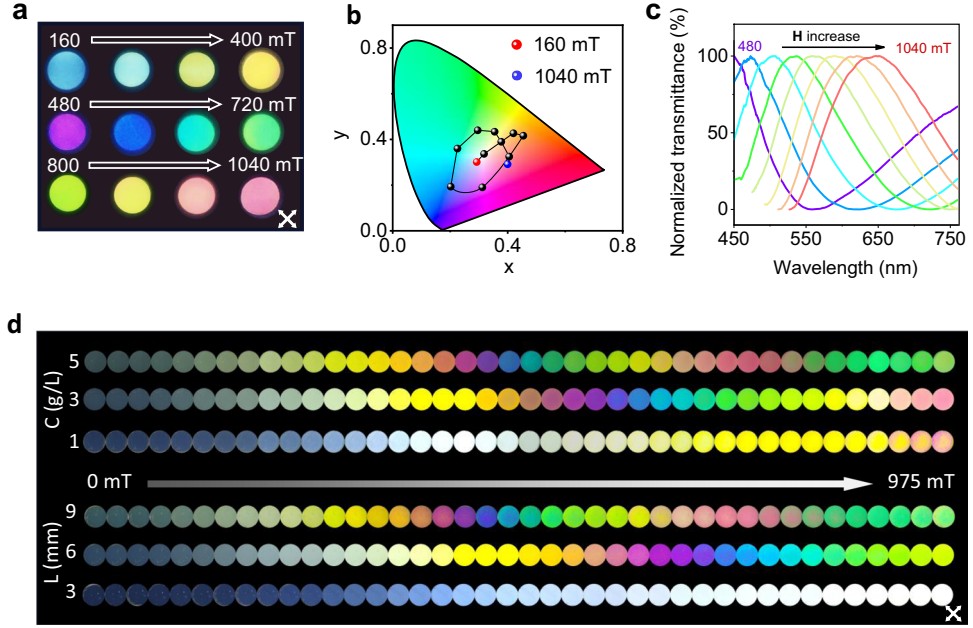

**Fig. 3 Transmitted interference colours of the MB-hydrogel. a-c** Polarized optical images (**a**), colour evolution (**b**) presented at a standard CIE-1931 colour space, and normalized transmission spectra (**c**) of the MB-hydrogels (No.13# to 24#). The hydrogels were cured under different magnetic fields in the range of 160 ~ 1040 mT with an interval of 80 mT. **d** The colour evolution of the MB-resin with different concentrations of 2D CTO materials (0.02, 0.06, 0.1 vol%) and thickness (3, 6, 9 mm) in the same magnetic-field range of 0 to 975 mT. The full scan of the snapshots can be seen in Supplementary Movie 3.

hydrogels include ones with 1) 2D disc-like materials of nontronite, polymersome, thulium ion ($Tm^{3+}$) and dysprosium ion ($Dy^{3+}$)-chelating bicelles; 2) 1D rod-like materials of fibrinogen, lead (Pb)-doped silica, polymer (poly (3-hydroxybutyric acid-co-3-hydroxyvaleric) acid (PHBV), agarose and 1,2-dipalmitoyl-sn-glycero-3-phosphocholine (DPPC). Meanwhile, the required magnetic field for the MB-hydrogel is one-order smaller than that for other systems, dropping from 10 T to 0.8 T, which can be supplied by an energy-free permanent magnet. We note that the required magnetic field to reach the saturate birefringence is even less than that of magnetically sensitive 0D sphere-like materials, such as iron oxide, whereas the strong absorption of iron oxide makes the hydrogel opaque. For CTO, the parent composite of 2D $TiO_2$ possesses unique optical properties such as large and anisotropic refractive index and excellent optical transmittance. In the meantime, the part substitution of Ti atoms by magnetic Co atoms can introduce the coupling between Co atom and neighboring Ti and O atoms, which forces Co spins to align within the Ti-O plane[39]. The enhanced magnetic anisotropy requires a smaller magnetic field to overcome the thermal relaxation and initiate the alignment of the CTO. As a result, CTO-based hydrogel concurrently exhibits high transparency, large birefringence, and desired engineerability.

The obtained large $\triangle n(H)$ and high transmittance (allowing long optical path of $L$) collectively give the MB- hydrogel with phase retardation of $\triangle n(H)L$ beyond 400 nm, which is the threshold value for producing transmitted interference colours according to Michel-Lévy chart (Supplementary Fig. 11). Therefore, the large magneto-phase-retardation permits the printing of the hydrogel with magnetically tunable colours. Due to the limited phase retardation (usually < 100 nm), previously reported magnetic hydrogels can only display the black-to-white switch and is not interference colourful. While for our MB-hydrogel, it is the transparent hydrogels with rich and tunable interference colours. Here, 12 MB-hydrogels (No. 13# to 24#) ($L = 6$ mm; $C = 0.02$ vol%) were fabricated in the fields of 160 to 1040 mT. In comparison with MB-hydrogels (No. 4# to 12#), the

thickness of MB-hydrogels (No.13# to 24#) is designed to be three times larger, in order to achieve the large phase retardation in the range beyond 400 nm. When back-lit with white light, each hydrogel gives its own interference colour (Fig. 3a). Colour indices at the standard CIE-1931 colour space quantify the field-colour correspondence (Fig. 3b). Note that once the hydrogel was cured and formed, the alignment of the 2D material is frozen inside the hydrogel, so its colour will be fixed (Supplementary Fig. 11). According to the Eq. (1), constructive interference occurs when the wavelength satisfies $\lambda_c = 2\triangle n(H)L$. Since $L$ keeps constant and $\triangle n$ monotonically increases with **H**, $\lambda_c$ is expected to redshift with **H**. This is consistent with the results in Fig. 3c as $\lambda_c$ shifts from 450 nm to 650 nm as **H** increases from 480 to 1048 mT. Besides **H**, modulation of $C$ and $L$ provides two more approaches to tuning the interference colours. For example, the interference colours of the MB-resin in the same **H** range (0 to 1.0 T) extend from the dark-white dominated first-order region ($\triangle nL < 500$ nm) at a concentration of 0.02 vol% to the second-order one at 0.06 vol% ($500 < \triangle nL < 1000$ nm), followed by entrance to the third-order one at 0.1 vol% ($1000 < \triangle nL < 1500$ nm) (top panel in Fig. 3d, Supplementary Movie 3). The similar colour evolutions for the MB-resins with different $L$ were also observable, which cover 1) dark-white dominated first-order region at 3 mm, 2) first- and second-order ones at 6 mm, and 3) the first-, second- and third-order ones at 9 mm (bottom panel in Fig. 3d). Diversified tuning methods and rich colour types permit the engineerable production of transmitted interference colours and thus the application of MB-hydrogels in the colour centred fields.

**Personalized MB-hydrogel device.** The superior magnetic and optical properties of MB- hydrogel allow the personalized optical applications based on such hydrogel. Several personalized proof-of-concept devices including phase retarder, gradient optical attenuator, magnetic see-through colour imager, and mechano-chromic and thermochromic indicators, which cannot be attainable in conventional magnetic hydrogels, have been successfully fabricated and demonstrated in our work. For the

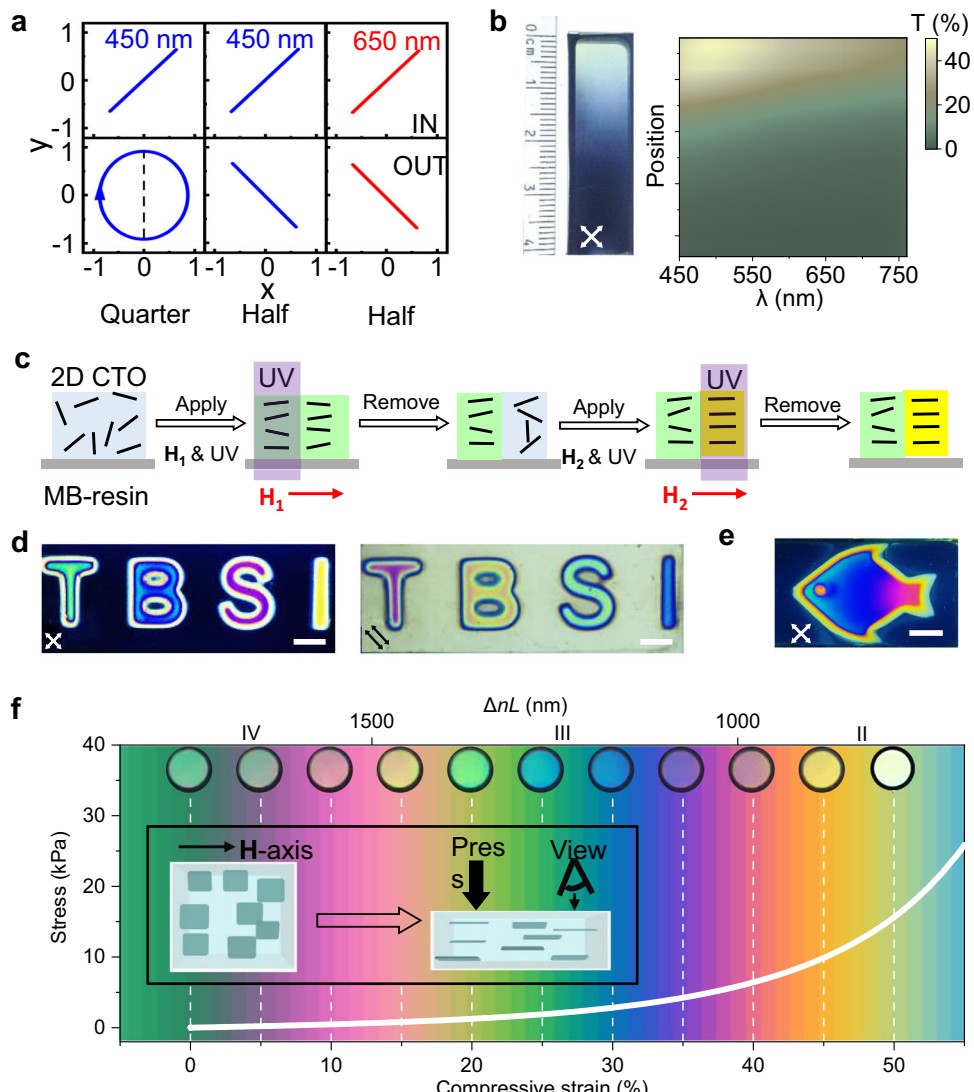

**Fig. 4 Optical applications of MB-hydrogels. a** MB-hydrogels (No. 25# to 27#) based quarter and half waveplates for monochromatic light of 450 nm and 650 nm. **b** MB-hydrogel-28 based gradient optical attenuator for visible light and its transmittance mapping as a function of location. The MB-hydrogel is solidified in a gradient magnetic field. **c** Procedure of magnetic see-through imaging based on the MB-hydrogel. **d** See-through images of "T", "B", "S", "I" letters on MB-hydrogel-29 with crossed polarizers (left) and parallel polarizers (right). Each colour is obtained by gelling hydrogel in a given magnetic field. Scale bar, 5 mm. **e** Boesemani-rainbowfish-like hydrogel. The MB-hydrogel-30 was synthesized in a gradient magnetic field. Scale bar, 10 mm. **f** Mechanochromic effect of the MB-hydrogel-31. Inset schematically shows the alignment change of 2D materials during pressing. The coloured background is from calculated Michel-Lévy chart in the phase-retardation range of 690 nm to 1840 nm. The direction of light propagation (or viewing direction) is along with the compressive force and normal to the **H**-axis. The real colours of MB- hydrogel-31 at different strains are displayed on the top and enclosed with the black circles. The full scan of the snapshots can be seen in Supplementary Movie 4.

MB-hydrogels with $\triangle nL$ in the first-order region (0 to 500 nm), they can be directly designed and used as true zero-order waveplates, such as a quarter wave plate for 450 nm (No. 25#) and two half-wave plates for 450 nm (No. 26#) and 650 nm (No. 27#) (Fig. 4a). Meantime, in the presence of crossed polarizers, a personalized gradient optical attenuator can be fabricated by using the MB-hydrogel-28 with a gradient phase retardation (Fig. 4b). The MB-hydrogel-28 was synthesized in a gradient magnetic field ranging from 0 to 400 mT. The quantified transmittance-wavelength-location mapping reveals the adjustable attenuation within the device region. For instance, the transmittance of the attenuator at 450 nm gradually decreases from 42.3% to 0% as the light spot moves downwards. It is worth noting that due to their thickness in the scale of few micrometres, conventional polymer- or crystal-based true zero-order waveplates usually require both high-precision surface polishing

techniques and microspheres as spacers to achieve arcsecond parallelism. Our MB-hydrogels are not subject to these limitations because of their thickness in the millimetre to centimetre scale, and can be readily synthesized in the bench-top lab.

When $\triangle nL$ goes beyond the first-order region (>500 nm), the generated interference colour allows its application in transmitted-colour centred fields. The process of magnetic see-through colour imaging is shown in Fig. 4c. The selected area of the MB-resin is exposed to UV light in the presence of a proper magnetic field. After solidification, the aligned magnetic 2D materials are frozen, giving rise to the consequential interference colour of the exposure area. Likewise, by exposing other areas with UV light and synchronously tuning the **H** strength, the expected colourful pattern can be lithographed. Despite the previous trials in a similar way, only the change in light contrast rather than colour was observed as a result of the strong optical absorption of iron oxide[33]. Alternatively, we

demonstrate the engineerable printing of patterned interference colours by using a transparent CTO as a functional unit. Four letters with patterns of "T", "B", "S", and "I" are printed on a thin MB-hydrogel-29 label ($C = 0.12$ vol%; $L = 1$ mm) (Fig. 4d). Each letter displays a unique interference colour, written by magnetically tuning and exposing to the focused UV light. The switch of polarizers from crossed (left) to parallel (right) ones results in the complementary colouration for each letter.

Moreover, inspired by many natural creatures with variable interference colours, such as Boesemani rainbowfish, we designed and printed a see-through rainbowfish image based on the MB-hydrogel-30 (Fig. 4e). The hydrogel was gelled in the gradient magnetic field again but had a six-times higher concentration of CTO ($C = 0.12$ vol%) compared to that of MB-hydrogel-28. The iridescent colour of the fish pattern corresponds to the gradient phase retardation. Meanwhile, once removing or changing polarizers, the MB-hydrogel-30 label becomes transparent or shows different appearances (Supplementary Fig. 12), suggesting the potential of a see-through colour image as a non-cloning, security-guaranteed label, if a randomly variable magnetic field is used, which was not possible using other conventional techniques.

In view of its superior elasticity, the MB-hydrogel can also serve as a mechano-chromic indicator. To ensure the uniform thickness, we designed a cubic MB-hydrogel-31, the strain-stress curve (Methods, Fig. 4f) reveals that it has Young's modulus of 597 kPa and bears up to 60% compressive strain. The inset of Fig. 4f schematically shows the alignment variation of 2D materials by a vertically exerted pressure. The force reduces vertical space and increases the electrostatic repulsion between negatively charged 2D CTO materials. Consequently, 2D materials are repulsed towards coplanar alignment with their planes normal to the compressive force, giving rise to the reduction (or increase) in both $L$ and $\triangle n$ along the vertical (or horizontal) direction. When viewing vertically, the taken snapshots on the top of Fig. 4f reveal that, with the deformation ratio from 0% to 20%, the colour gradually turns from the initial forth-order green (0%) to purple (10%), and then to the third-order green (20%). With further pressing, the colour becomes second-order yellow (45%). Switching the view direction from the vertical one to the horizontal one, the tendency is reversed accordingly (Supplementary Movie 4). The developed MB hydrogel also shows its response to the thermal stimulus. To avoid the impact of the thermally induced water loss, we fabricated an encapsulated MB-hydrogel (Supplementary Fig. 13). When the temperature rises from 20 °C to 76 °C, the colour of the hydrogel sandwiched between two crossed polarizers changes accordingly and evolves from pink to light green. Therefore, the MB-hydrogel has both mechanochromic effect and thermochromic effect.

## Discussion

We have fabricated a transparent and optical-anisotropy tuneable hydrogel by embedding magnetically aligned 2D materials into a polymer matrix. The alignment order of 2D materials has been controlled by exerting a magnetic field during hydrogelation. High transparency, good uniformity, large and widely tuneable anisotropy cooperatively offer the hydrogel capability of optical modulation, including the manipulation of light polarization, intensity, and colour. These unique features open the door for the direct use of anisotropic hydrogels as optical elements, such as phase retarders, optical filters, gradient optical attenuators, magnetic see-through imagers and mechano-chromic indicators. These demonstrations on the application of anisotropic transparent hydrogel in optics suggest its great potential for customized optical elements, vision enhancement devices, unclonable anti-counterfeiting labels, and other anisotropy-related fields.

## Methods

**Synthesis of the MB-hydrogel**. The MB-hydrogel contains the polymerized poly(ethylene glycol) diacrylate (PEGDA) as a matrix, and magnetic 2D CTO materials as functional units. The process to synthesize the 2D CTO materials includes three stages. In stage I, layered parent compound $K_{0.8}Ti_{1.67}Li_{0.13}Co_{0.2}O_4$ was prepared via high-temperature crystallization. $TiO_2$ (0.25 mol, 20 g), CoO (0.03 mol, 2.25 g), $K_2CO_3$ (0.06 mol, 5.94 g), and $Li_2CO_3$ (0.01 mol, 0.67 g), all from Shanghai Aladdin Biochemical Co., Ltd., China, were mixed in a stoichiometric ratio by grinding in a corundum crucible and annealed at 1000 °C for 5 h. In stage II, the expanded $H_{0.93}Ti_{1.67}Co_{0.2}O_4$ was prepared via the protonic exchange of $Li^+$ and $K^+$ with $H^+$ (Supplementary Fig. 1). 1 g parent layered materials was mixed with 200 ml HCl (1 M) and agitated for 4 days using a magnetic stirrer to allow sufficient ion exchange. In stage III, the final product of unilamellar or few-layer $Ti_{0.83}Co_{0.2}O_2^{0.47-}$ (CTO) in water was prepared via the ionic exchange of inter-layered $H^+$ with tetrabutylammonium ions ($TBA^+$) (Supplementary Fig. 2). The powder of $H_{0.93}Ti_{1.67}Co_{0.2}O_4$ powder was soaked and softly shaking in TBAOH aqueous solution ($H^+:TBA^+ = 1:1$ in molar ratio) for 5 h. Whereafter, adding the PEGDA (monomer, 5 wt%, average molecular weight of 700 g/mol, Sigma-Aldrich, USA) and 2-hydroxy-4'-(2-hydroxyethoxy)-2-methylpropiophenone (Irgacure 2959, photo-initiator, 1 wt%, molecular weight of 224.25 g/mol, Aladdin, USA) into the CTO suspension gave the UV-curable mixture of the MB-resin. As-fabricated MB-resin was then injected into the customized containers, which were then placed between two poles of an electromagnet or permanent magnet to align 2D materials. The magnetic field can be tuned in the range of 0 −1 Tesla. Finally, MB-hydrogels were obtained by solidification of the MB-resin by exposure to UV light with a peak wavelength at 365 nm and a power of ~1 W/cm². Curing time is dependent on the concentration of 2D CTO and was in the range of 5 ~100 s. The birefringence of MB-hydrogel can be controlled by tuning the strength of the applied magnetic field during hydrogelation. PEGDA and Irgacure 2959 with low concentrations in the MB-resin are chosen in our work as they show a fast polymerization reaction rate, no additional birefringence, and negligible influence on the viscosity and alignment of the 2D materials.

**Encapsulation of the hydrogel**. After the preparation of MB-hydrogel, waterproof glue and packaging substrates such as glass or sealing plastic were immediately used to isolate the hydrogel from the air, so that the water molecules in the hydrogel are physically isolated. As shown in Supplementary Fig. 14, the bare hydrogel (unencapsulated) experiences the obvious shrinkage after 6 h in air, indicating severe and quick water loss. As a consequence, the colour changes accordingly due to the reduced optical path and varied alignment order of CTO by the water loss. As a sharp contrast, for the encapsulated hydrogel, no obvious colour change was observed and the hydrogel still keeps transparent even after one week in the air, confirming the effectiveness of the physical isolation method in keeping high transparency and preventing the water loss.

**Characterization of the MB-hydrogel**. The polarization was monitored with a polarimeter (PAX1000VIS, Thorlabs, Inc. USA), and colour spectra and intensity of modulated light were collected using an optical spectrometer (USB2000 UV-VIS, Ocean Optics, Inc. USA). For polarized optical images, a surface light source comprising a white light-emitting diode was used as the backlight. The strain-stress test for the MB-hydrogel was characterized in a Model 5943 Materials Testing System (Instron, Illinois Tool Works Inc. USA). The compression rate was set at 6 mm/min during experiments. The interaction between CTO and hydrogel was characterized by using XPS (Model: ESCALAB 250Xi, Thermo Fisher, England). Morphology of freeze-dried hydrogels was characterized by using SEM at 5 keV (Hitachi SU8010, Japan).

**Statistics and Reproducibility**. Consistent results for all representative experiments (including the measurement of optical anisotropy (Fig. 2b,d) and mechano-chromatic effect (Fig. 4f) are repeated via $n \geq 3$ independent experiments.

## Data availability

NA. Source data are provided with this paper.

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

## Acknowledgements

We acknowledge support by the National Natural Science Foundation of China (No. 51920105002), the Guangdong Innovative and Entrepreneurial Research Team Program (No. 2017ZT07C341), the Shenzhen Basic Research Project (Nos. JCYJ20190809180605522, and WDZC20200819095319002), the National Key R&D Program (2018YFA0307200), and the Bureau of Industry and Information Technology of Shenzhen for the "2017 Graphene Manufacturing Innovation Center Project" (No. 201901171523).

## Author contributions

B.D., P.Z., H.-M.C. and B.L. designed and directed the project. B.D., Y.P. and Z.H. synthesised the materials and performed materials-related characterisation. B.D., P.Z., T.L., L.D., H.X. and B.L. carried out the optical and mechanic characterisation and analysis. B.D., P.Z. and B.L. performed the theoretical analysis. B.D., P.Z., Y.L., T.L., Z.H., Q.Y., H.-M.C. and B.L., analysed the data and co-wrote the paper with feedbacks from other authors.

## Competing interests

The authors declare the following competing interests: Patents related to this research have been filed by Tsinghua-Berkeley Shenzhen Institute, Tsinghua University. The University's policy is to share financial rewards from the exploitation of patents with the inventors.
