## [Peer Review File · Nature Communications]

A transparent hydrogel with engineerable interference coloursReviewers' comments:

Reviewer #1 (Remarks to the Author):

In this article, the authors fabricated a transparent and optical-anisotropy tunable hydrogel by embedding magnetically aligned 2D materials into hydrogel. This engineered hydrogel possesses high transparency, good uniformity, large and widely tunable anisotropy, which help the demonstration of several applications at the end of this article. This study can bring interest to researchers in the field of materials and optical devices. However, the innovation of this work can not be clearly caught at the current version since the method to synthesize the 2D CTO materials was already presented in existing literature, as well as no new method was introduced to prepare the hydrogel here. The authors are also suggested to improve the work by considering the following comments.

(i) The interface between the 2D CTO and the hydrogel should be characterized if possible, and its effect on the durability of the device need to be investigated or discussed.

(ii) It's very interesting to show that the MB-hydrogel can serve as a mechano-chromic indicator in this article. Other functions to tune the colour are necessary to present. For example, is magnetic field useful to tune the colour?

(iii) It's a big problem for hydrogel to maintain the water content for a considerable time. Is there any method to keep high transparency with the current water maintaining approaches for hydrogel?

Reviewer #2 (Remarks to the Author):

In this paper, the authors invent a transparent magneto-birefringence hydrogel with large and finely engineerable optical anisotropy. High transparency, sensitive magnetic response, large and tuneable optical anisotropy cooperatively permit the magnetic patterning of interference colours in the hydrogel. This finding provides an entry point for applying hydrogel in optical anisotropy and colour centred fields.

Scanning or transmission electron microscope analysis of the product is required to prove CTO's ordered arrangement.

Strength data of materials, such as tension, compression, etc. needs be provided.

Point-to-Point Response to Reviewers

List of changes (blue in the revised manuscript and SI).

- (1) Added 7 new figures in the revised SI (Supplementary Figures 3-5, 9, 12, 14, 15).
- (2) Added some sentences in the revised manuscript (Pages 2, 5-7, 10, 13-18, 24, 25).
- (3) Added 1 new reference in the revised manuscript (Ref. 35)

Reviewer #1:

Comment: In this article, the authors fabricated a transparent and optical-anisotropy tunable hydrogel by embedding magnetically aligned 2D materials into hydrogel. This engineered hydrogel possesses high transparency, good uniformity, large and widely tunable anisotropy, which help the demonstration of several applications at the end of this article. This study can bring interest to researchers in the field of materials and optical devices.

Response: We thank the Reviewer very much for the overall positive recommendations. We also appreciate the reviewer by writing that this study “can bring interest to researchers in the field of materials and optical devices”.

Comment 1: However, the innovation of this work cannot be clearly caught at the current version since the method to synthesize the 2D CTO materials was already presented in existing literature, as well as no new method was introduced to prepare the hydrogel here.

Response: We apologize that we did not write the innovation of the work clearly in the previous version. We fully agree with the reviewer that the method to synthesize 2D CTO materials and prepare hydrogel is not new. Actually, the key innovation of this work is not the material preparation or hydrogel preparation method itself, but the first report of a hydrogel with tunable interference colours as well as their new-working-principle optical device applications. We have followed your comments, re-organized the words to help our readers clearly catch the innovation and originality of this work. Please see below for details.

1. Fabrication of the first hydrogel with tunable interference colours.

Contrary to the colours from dyes or pigments, interference colours show unique vivid, metallic, wide colour gamut and non-photobleaching advantages. Therefore, it is highly desired to introduce interference colour functions into hydrogels. Unfortunately, current reported transparent hydrogels can only display the black-

to-white switch and is not interference colourful. Here, we fabricated the first transparent hydrogel with tunable interference colours, whose colour can be continuously switched from red to green and further to blue (Figure 3 in the MS). To our best knowledge, such hydrogel with tunable interference colours was not reported before, which provides a unique platform for optical applications of hydrogel.

2. **Demonstration of new-concept and personalized optical applications based on the MB-hydrogel.** Several personalized new-working-principle optical devices including flexible optical phase retarder, gradient optical attenuator, magnetic see-through colour image, and mechano-chromic/thermochromic indicator, which cannot be attainable in conventional hydrogels, have been demonstrated in our work based on the MB-hydrogel. Among them, the see-through colour image can also be used as a non-cloning, security-guaranteed label, if a randomly variable magnetic field is used in hydrogel synthesis process, which is not possible by other conventional techniques. The realization of these proof-of-concept devices also supports the novelty and uniqueness of our MB-hydrogel in optics.

Changes to the revised manuscript. We have added the discussions in the main text to help our readers clearly catch the innovation of the work. On Page 2, “Contrary to the colours from dyes or pigments ... and is not interference colourful.” On Page 10, “a transparent hydrogel ... to the value of < 1 tesla”. On Page 12, “Therefore, the large magneto-phase-retardation ... the first transparent hydrogels with the rich and tunable interference colours.” On Pages 14, 15, “The superior magnetic and optic property ... will be demonstrated subsequently.” On Page 17, “suggesting the potential of see-through colour image ... not possible using other conventional techniques.”

Comment 2: The interface between the 2D CTO and the hydrogel should be characterized if possible, and its effect on the durability of the device need to be investigated or discussed.

Response: This is a very inspiring comment. We have followed your suggestion and characterized the interface interaction between the 2D CTO and the polymer network in hydrogel by XPS. Based on XPS measurements (Figure R1), we detected a peak shift of Ti 2p_{3/2} from 458.47 eV (in pure 2D CTO flake, which is in good agreement with the standard value of TiO₂, see: <http://srdata.nist.gov/xps/>) to a lower binding energy of 457.44 (in polymer-PEGDA modified 2D CTO flakes). Such peak shift indicates interface interaction between CTO and polymers, presumably due to the formation of hydrogen bond between hydrogen in polymer network and oxygen in CTO, making a change from Ti⁺⁴ to Ti^{+(4-δ)}. When we compare Ti-O-H (in the case of hydrogel) with

Ti-O (in the case of pure TiO₂), the O in Ti-O-H can withdraw electrons from both Ti and H due to its large electronegativity, while O in Ti-O can only withdraw electrons from Ti. This leads to a less decrease of 3d/4s-orbital electron density of Ti in Ti-O-H than in Ti-O. Such less decrease of Ti 3d/4s electron density will result in an outward shift of Ti 2p-orbital electrons in Ti-O-H case than in Ti-O case, due to the so-called screening effect. This difference leads to a lower binding energy for Ti 2p in Ti-O-H (hydrogel) than in Ti-O (pure TiO₂) case, consistent with our XPS measurements (Figure R1). This result is also in accordance with literature on the similar analysis of Metal-O-H bonding (see for example, Kim, et al. Applied Surface Science, 1999, 152, 35-54).

We also followed your comments and investigated such interface effect on the durability of hydrogels by testing the mechanical properties of hydrogels without and with CTO flakes. We found that addition of CTO remarkably improves the mechanical performance of hydrogel as evidenced by stress-strain curves during compression or tension (Figure R2) as well as the mechanical durability (Figure R3). For instance, for tension-strain correspondence, the stress of hydrogel at 20 kPa increases from 10 kPa to 15 kPa after adding CTO, giving rise to an increase by 50 %. For the durability, the retention rate of hydrogel without CTO significantly drops to 60% after cycling for 14 times. Meanwhile, the hydrogel was fractured into several pieces, indicating its poor durability. While after adding CTO, the hydrogel shows much improved durability, as evidenced by the retention rate of > 90% after cycling for 50 times.

Changes to the revised manuscript. On Pages 5, 6, “Based on XPS measurements (Supplementary Fig. 3) ... Metal-O-H bonding, where M is the typical metal.” and “If the CTO-hydrogel interface is dominated by the H-bond... as evidenced by the retention rate of > 90%, even after 50-time cycling.” On Page 25, “The interaction between CTO ... (Model: ESCALAB 250Xi, Thermo Fisher, England).”

Figure R1| Characterization of interface interaction between 2D CTO and polymer network in the hydrogel. XPS spectra of Ti 2p_{3/2} of two filtered films based

on pure 2D CTO flakes (blue curve) and polymer-PEGDA modified 2D CTO flakes (red curve), the Ti 2p_{3/2} peaks located at 458.47 eV and 457.44 eV, respectively. Inset shows the XPS spectra of C 1s peaks at 284.8 eV used for calibration. This figure was added as **Supplementary Fig. 3** in the revised Supplementary Information.

Figure R2| Mechanical properties of hydrogels without and with CTO. a, b, Stress-strain curves for tension (a) and compression (b). This figure was added as **Supplementary Fig. 4** in the revised Supplementary Information.

Figure R3| Cycling performance of hydrogel without and with CTO. a, Retention rates of hydrogels without CTO (black curve) and with CTO (red curve). **b,** Strain (top panel), compressions of hydrogels with CTO (middle panel) and without CTO (bottom

panel) for the first 14 cycling periods. This figure was added as **Supplementary Fig. 5** in the revised Supplementary Information.

Comment 3. It's very interesting to show that the MB-hydrogel can serve as a mechano-chromic indicator in this article. Other functions to tune the colour are necessary to present. For example, is magnetic field useful to tune the colour?

Response: Thank you very much for writing that our mechano-chromic indicator is “very interesting”.

Inspired by your question, we find one more function to tune the colour of hydrogel besides the mechanic force, i.e., temperature. As shown in Figure R4, when the temperature rises from 20 to 76 °C, colours of the hydrogel change accordingly and evolve from pink to light green. Therefore, the MB-hydrogel has both mechano-chromic effect and thermo-chromic effect.

About the colour response of hydrogel to magnetic stimuli, we can achieve different colours of the hydrogel by varying the strength of magnetic field **during** the curing process. Under given concentration of CTO, the colour of hydrogel shows one-to-one correspondence with the strength of magnetic field. Therefore, magnetic field can be used to tune the colour of hydrogel during UV curing. This has been verified by images shown in Fig. 3 in manuscript. To check whether the magnetic field can change the colour of MB-hydrogel **after** solidification (hydrogel already formed), we fabricated two additional coloured MB-hydrogels with different concentrations of CTO. We found that once the hydrogel was cured and formed, the alignment of the 2D material is fixed inside the hydrogel, so its colour is fixed and cannot be tuned by magnetic field. As shown in Figure R5, the hydrogel shows negligible colour change by switching the external magnetic field from 0 to 1 T. This result supports the good stability of the hydrogel once it formed, and is also a basic requirement that the MB-hydrogel can record and store the optical or colour information after removal of magnetic field.

Changes to the revised manuscript. On **Page 14**, “Note that once the hydrogel was cured and formed ... so its colour will be fixed (Supplementary Fig. 12).” On **Page 18**, “The developed MB hydrogel also shows its response to the thermal stimulus ... has both mechano-chromic effect and thermo-chromic effect.”

Figure R4| Thermochromic effect of the MB-hydrogel. **a**, A photo of a MB-hydrogel with transparent substrates encapsulate on both sides to avoid water loss during heating/cooling process. **b**, Colour evolution of the MB-hydrogel from pink to light green with the increase of temperature. The images were taken under two crossed polarizers as indicated. This figure was added as **Supplementary Fig. 14** in the revised Supplementary Information.

Figure. R5| Polarized optical images of MB-hydrogels in different magnetic fields. Two already-formed MB-hydrogels with different CTO concentrations in the absence (top panel) and presence (bottom panel) of magnetic field of 1 T. No colour change is seen, indicating once polymer is cured and hydrogel is formed, its color will be fixed and keep stable under different magnetic field. This figure was added as **Supplementary Fig. 12** in the revised Supplementary Information.

Comment 4. It's a big problem for hydrogel to maintain the water content for a considerable time. Is there any method to keep high transparency with the current water maintaining approaches for hydrogel?

Response: Water loss is indeed a big problem in hydrogel and some approaches have been developed to maintain water content currently. Here we followed a simple

physical isolation method which is found to be effective in keeping high transparency and maintaining water content in our MB-hydrogel. Specifically, we encapsulated the MB-hydrogel with waterproof glue and packaging substrates such as glass or sealing plastic to isolate the hydrogel from air. Therefore, the water molecules in the hydrogel are physically isolated from environment and sealed by the packaging substrates. As compared in Figure R6, the bare hydrogel (unencapsulated) experiences obvious shrinkage after 6 hours in air, indicating severe and quick water loss. As a consequence, its colour changes due to the reduced optical path and varied alignment order of CTO by the water loss. As a sharp contrast, for the encapsulated hydrogel, no obvious colour change was observed and the hydrogel still keeps transparent even after one week in air, confirming the effectiveness of physical isolation method in preventing water loss.”

Changes to the revised manuscript. On Page 24, “After the preparation of MB-hydrogel ... keeping high transparency and preventing the water loss.”

Figure R6| Stability of the MB-hydrogel without and with encapsulation. Bottom encapsulated hydrogel was sealed by using waterproof glue and packaging substrates. This figure was added as **Supplementary Fig. 15** in the revised Supplementary Information.

Reviewer #2:

Comment: In this paper, the authors invent a transparent magneto-birefringence hydrogel with large and finely engineerable optical anisotropy. High transparency, sensitive magnetic response, large and tuneable optical anisotropy cooperatively permit the magnetic patterning of interference colours in the hydrogel. This finding provides an entry point for applying hydrogel in optical anisotropy and colour centred fields.

Response: Thank you very much for your overall positive recommendations. We are grateful for the reviewer's comment that our finding "provides an entry point for applying hydrogel in optical anisotropy and colour centred fields".

Comment 1: Scanning or transmission electron microscope analysis of the product is required to prove CTO's ordered arrangement.

Response: We have followed your suggestion and studied CTO's ordered arrangement by SEM. Since SEM analysis needs to be done in vacuum where water is not allowed, to prepare the sample we used the freeze-drying method to remove water from the MB-hydrogel while reserving the original alignment order of CTO. We characterized freeze-dried hydrogels cured with and without magnetic field for comparison. As shown in Figure R7, the SEM image of freeze-dried hydrogel without a magnetic field shows random orientation of the 2D CTO in its cross-section. While for the sample cured with a magnetic field of 1 T, most of CTO materials oriented along with the magnetic field direction. Therefore, together with the results of the optical method shown in Fig. 1b-d of manuscript, we conclude the parallel alignment of 2D CTO with **H**-axis.

Changes to the revised manuscript. On Page 7, "Moreover, as shown in scanning electron microscopy (SEM) images (Supplementary Fig. 9) ... confirm the parallel alignment of 2D CTO with **H**-axis." On Page 25, "Morphology of freeze-dried hydrogels was characterized by using SEM at 5 keV (Hitachi SU8010, Japan)."

Figure R7| Study of ordered arrangement of CTO in MB-hydrogel made without or with magnetic field. a-d SEM images of freeze-dried MB-hydrogels cured without (a,b) and with (c,d) an external magnetic field of 1 T. This figure was added as **Supplementary Fig. 9** in the revised Supplementary Information.

Comment 2: Strength data of materials, such as tension, compression, etc. needs be provided.

Response: We have followed your nice suggestions and performed additional experiments to measure the mechanic durability and tension/compression-strain correspondence of the MB-hydrogel. These experimental results are shown in Figures R8 and R9. From the comparison, it can be seen that the introduction of CTO improves both the tension/compression performance and the durability of the hydrogel.

Changes to the revised manuscript. On **Page 5**, “If the CTO-hydrogel interface is dominated by the H-bond ... even after 50-time cycling.”

Figure R8| Cycling performance. **a**, Retention rates of hydrogels without CTO (black curve) and with CTO (red curve). **b**, Strain (top panel), compressions of hydrogels with CTO (middle panel) and without CTO (bottom panel) for the first 14 cycling periods. This figure was added as **Supplementary Fig. 5** in the revised Supplementary Information.

Figure R9| Mechanical properties of hydrogels without and with CTO. **a**, **b**, Stress-strain curves for tension (**a**) and compression (**b**). This figure was added as **Supplementary Fig. 4** in the revised Supplementary Information.

REVIEWERS' COMMENTS

Reviewer #1 (Remarks to the Author):

This revised manuscript improved a lot and answered my questions carefully. It is recommended to be accepted.

Reviewer #2 (Remarks to the Author):

This paper has been modified, and the corresponding data has been supplemented. I recommend this article to be published in Nature Communications.

Response to Reviewers' Comments

Reviewer #1: This revised manuscript improved a lot and answered my questions carefully. It is recommended to be accepted.

Response: We are so pleased to receive your agreement on our response in the last version and thank you very much for your positive recommendation.

Reviewer #2: This paper has been modified, and the corresponding data has been supplemented. I recommend this article to be published in Nature Communications.

Response: We feel happy to receive your satisfaction about the supplied data and thank you very much for your positive recommendation.